# Effects of Soluble Dextrin Fiber from Potato Starch on Body Weight and Associated Gut Dysbiosis Are Evident in Western Diet-Fed Mice but Not in Overweight/Obese Children

**DOI:** 10.3390/nu16070917

**Published:** 2024-03-22

**Authors:** Paweł Czarnowski, Aneta Bałabas, Zbigniew Kułaga, Maria Kulecka, Krzysztof Goryca, Kazimiera Pyśniak, Katarzyna Unrug-Bielawska, Anna Kluska, Katarzyna Bagińska-Drabiuk, Maria Głowienka-Stodolak, Magdalena Piątkowska, Michalina Dąbrowska, Natalia Żeber-Lubecka, Aldona Wierzbicka-Rucińska, Aneta Kotowska, Sebastian Więckowski, Michał Mikula, Janusz Kapuśniak, Piotr Socha, Jerzy Ostrowski

**Affiliations:** 1Department of Genetics, Maria Sklodowska-Curie National Research Institute of Oncology, 02-781 Warsaw, Poland; pawel.czarnowski@nio.gov.pl (P.C.); aneta.balabas@nio.gov.pl (A.B.); mkulecka@cmkp.edu.pl (M.K.); kgoryca@gmail.com (K.G.); kazimiera.pysniak@nio.gov.pl (K.P.); katarzyna.unrug-bielawska@nio.gov.pl (K.U.-B.); anna.kluska@pib-nio.pl (A.K.); katarzyna.baginska@pib-nio.pl (K.B.-D.); maria.glowienka@pib-nio.pl (M.G.-S.); magdalena.piatkowska@pib-nio.pl (M.P.); michalina.dabrawska@pib-nio.pl (M.D.); natalia.zeber-lubecka@cmkp.edu.pl (N.Ż.-L.); michal.mikula@nio.gov.pl (M.M.); 2Department of Biochemistry, Radioimmunology and Experimental Medicine, Children’s Memorial Health Institute, 04-730 Warsaw, Poland; a.wierzbicka-rucinska@ipczd.pl; 3Public Health Department, Children’s Memorial Health Institute, 04-730 Warsaw, Poland; z.kulaga@ipczd.pl (Z.K.); a.kotowska@ipczd.pl (A.K.); 4Department of Gastroenterology, Hepatology and Clinical Oncology, Centre of Postgraduate Medical Education, 02-781 Warsaw, Poland; 5Genomic Core Facility, Centre of New Technologies, University of Warsaw, 02-097 Warsaw, Poland; 6Department of Gastroenterology, Hepatology and Eating Disorders, Children’s Memorial Health Institute, 04-730 Warsaw, Poland; s.wieckowski@ipczd.pl (S.W.); p.socha@ipczd.pl (P.S.); 7Department of Dietetics and Food Studies, Faculty of Science and Technology, Jan Dlugosz University, 42-200 Czestochowa, Poland; j.kapusniak@ujd.edu.pl

**Keywords:** soluble dextrin fiber, obesity, animal and human studies, gut microbiome, gut metabolites

## Abstract

Background: The study investigated the impact of starch degradation products (SDexF) as prebiotics on obesity management in mice and overweight/obese children. Methods: A total of 48 mice on a normal diet (ND) and 48 on a Western diet (WD) were divided into subgroups with or without 5% SDexF supplementation for 28 weeks. In a human study, 100 overweight/obese children were randomly assigned to prebiotic and control groups, consuming fruit and vegetable mousse with or without 10 g of SDexF for 24 weeks. Stool samples were analyzed for microbiota using 16S rRNA gene sequencing, and short-chain fatty acids (SCFA) and amino acids (AA) were assessed. Results: Results showed SDexF slowed weight gain in female mice on both diets but only temporarily in males. It altered bacterial diversity and specific taxa abundances in mouse feces. In humans, SDexF did not influence weight loss or gut microbiota composition, showing minimal changes in individual taxa. The anti-obesity effect observed in mice with WD-induced obesity was not replicated in children undergoing a weight-loss program. Conclusions: SDexF exhibited sex-specific effects in mice but did not impact weight loss or microbiota composition in overweight/obese children.

## 1. Introduction

Obesity is responsible for the development of various human pathologies, including cancer, metabolic syndrome, and cardiovascular disorders [1,2,3]. In 2016, 1.9 billion adults worldwide were overweight, 650 million of whom were obese, and more than 340 million children and adolescents aged 5–19 years were affected by overweight or obesity. In 2020, 39 million children younger than 5 years were overweight or obese [4]. Consequently, obesity is a public health burden with profound impacts on population morbidity, mortality, and healthcare costs [5].

Obesity results from an imbalance between energy intake and expenditure. The main reason for the obesity epidemic is related to the global proliferation of the so-called “Western lifestyle”, which comprises a Western diet (WD) and largely sedentary daily routines. A WD may refer to different eating patterns [6,7], but all of them are characterized by a combination of high-calorie, high-fat, high-refined carbohydrate, and low-digestible fiber consumption. Therefore, it is unsurprising that lifestyle modifications remain the primary goal in the prevention and treatment of obesity and obesity-related disorders [8].

The metabolic functions of the host are partly impacted by the gut microbiota that harvests nutrients and energy from the diet through anaerobic fermentation of non-digestible and non-hydrolysable dietary carbohydrates in the colon [9,10]. Both an unhealthy diet and low physical activity modulate the functional composition and metabolic activities of the gut microbiota, leading to the obesity-related imbalance and local distribution shifts of the gut microflora, called dysbiosis [11]. Consequently, prebiotic- and/or probiotic-based modifications of the gut microbiota are considered helpful to prevent and treat obesity [8].

According to the Food and Agriculture Organization of the United Nations and World Health Organization (WHO), prebiotics are “non-digestible food ingredients that beneficially affect the host by selectively stimulating the growth and/or activity of one or a limited number of bacterial species already established in the colon, and thus improve the host health” [12]. In addition to their modulatory effects on the functional composition of the gut microbiota, prebiotics, including inulin, resistant starch, and β-glucans, may control caloric intake by delaying gastric emptying and reducing appetite. The amounts of prebiotics are reduced in a WD, and food rich in prebiotics has beneficial effects on obesity and obesity-related disorders [8].

Among the different types of prebiotics, dietary fibers composed of carbohydrates are resistant to hydrolysis by human digestive enzymes in the small intestine but can be fermented by the colonic microbiota [13]. Following clinical trials verifying the beneficial effects of resistant dextrin, dextrin has been used in products, such as Fibersol-2 and Nutriose [14]. Thermolysis of potato starch in the presence of hydrochloric, citric, and tartaric acids under controlled conditions is one of the methods used to produce soluble but non-digestible dietary dextrin with the expected prebiotic properties [15,16,17,18].

The purpose of our study was to assess whether soluble dextrin fiber (SDexF) produced from potato starch can support the weight loss in children with overweight/obesity who underwent an obesity intervention and/or may reduce the risk of weight gain in mice, and to determine whether these possible effects are accompanied by changes in the composition of the gut microbiome and metabolites.

## 2. Materials and Methods

SDexF was prepared according to patent PL220965, however, after increasing the scale to semi-industrial. Thus, potato starch (10 kg) was poured into the mixer of the testing device. The mixer was started at a rotational speed of 100 rpm. Then, using a spray nozzle, the weighed starch was sprayed with 1677 mL of a 0.5% (*w*/*w*) food grade hydrochloric acid solution, and then with the same amount of food grade aqueous citric acid solution (0.5% *w*/*w*), so as to obtain final acid concentrations of 0.1% of starch dry weight. During spraying, the dedusting system was activated. In the next stage (with continuous mixing at 100 rpm), the acidified starch was dried simultaneously with air (temp. 105 °C) and thermal oil (temp. 110 °C) until the moisture content was lower than 5% (the water content was controlled using moisture analyzer after every 1 h of drying). After the drying stage, the hot air supply and the dedusting system were turned off. Then, the sample in the mixer was heated at 130 °C for 4 h, using only thermal oil. After this time, the SDexF was cooled, sieved through laboratory sieves made of stainless steel to remove burnt formed during heating, and then washed with several portions of 80% ethyl alcohol to remove low-molecular products of starch thermolysis and citric acid residues. In order to remove the remains of ethyl alcohol, the washed SDexF was dried on glass trays at 80 °C for 24 h (dryer with gravity air circulation, with a chamber and shelves made of stainless steel).

The research material consisted of vegetable and fruit mousses without and with the addition of an SDexF produced by Tymbark MWS Sp. z o. o. (Tymbark, Poland). Three flavors of mousses were used for testing including apple-carrot-quince, apple-peach-parsnip, and apple-cherry-carrot. Mousses were prepared in 100 g tube packages. The SDexF content in each tube was 5 g. Children consumed 2 tubes every day. This means that the total daily dose of SDexF administered in the form of mousse was 10 g.

### 2.1. Animal Study

C57BL6/W mice were born in a specific pathogen-free (SPF) core facility at the National Research Institute of Oncology, Warsaw and maintained under standard humidity (55% ± 10%) and temperature (21 ± 2 °C) conditions in climate-controlled rooms under a 12-h light cycle. Animals were tested for the presence of viruses, bacteria, and parasites according to the recommendations of the Federation of European Laboratory Animal Science Associations.

The normal diet (ND) and Western diet (WD) were supplied by the manufacturer as a well-granulated powder. After adding 20 mL of water to 100 g of powdered feed, a paste formulation was obtained. SDexF-supplemented diets were prepared by mixing 95 g of powdered feed with 5 g of SDexF and 20 mL of water. Animals had unrestricted access to water and food throughout the experiment.

Ninety-six 12-week-old C57BL6/W mice of both sexes fed an ND (containing 10% fat, 30% protein, and 51% carbohydrate) (Labofeed H, Feed Factory Morawski, Kcynia, Poland) were allowed to adapt to a non-SPF environment for 2 weeks. Animals were then randomly divided into four groups, each comprising 24 mice (12 males and 12 females), and fed an ND (containing 4.2 g of fat, 22 g of protein, and 37 g of carbohydrate in 100 g of dry weight of feed, with a metabolic energy content of 12.8 MJ), an ND supplemented with 5% SDexF, a WD (containing 21 g of fat, 19.8 g of protein, 50 g of carbohydrate, and 0.15 g of cholesterol in 100 g of dry weight of feed, with a metabolic energy content of 18.4 MJ) (Feed Factory, Morawski, Poland), and a WD supplemented with 5% SDexF, respectively. The body weight of the mice was measured weekly to an accuracy of 0.1 g for 28 weeks (Figure 1). Two mice fed a WD without SDexF were terminated due to significant body weight loss and overall bad condition on week 3 (female) and week 9 (male). At the end of the experiment, the mice were anesthetized using 5% isoflurane and sacrificed by cervical dislocation, and blood samples and livers were acquired immediately. Stool samples were collected from each mouse before and at the end of the experiment, and stored at −80 °C until use.

The experimental procedure was approved by the National Ethics Committee for Animal Testing (decision: 1/2022 on 14 January 2022), and the studies were conducted in accordance with the Directives of the European Parliament and of the Council (2010/63/EU) and the Polish regulations on the protection of animals used for scientific and educational purposes (Journal of Laws 2021, items 1331 and 2338).

### 2.2. Serum Biochemical Measurements

Levels of serum cholesterol, low-density lipoprotein-cholesterol, triglyceride, alanine aminotransferase, aspartate aminotransferase, and alkaline phosphatase were measured using an automated clinical chemistry analyzer (Spotchem EZ SP-4430; ARKRAY, Kyoto, Japan) and the corresponding test strips.

### 2.3. Histological Analysis

A freshly collected liver tissue sample measuring 1 cm^2^ was washed with saline, immediately fixed in 4% paraformaldehyde solution, and stained with hematoxylin and eosin to histologically evaluate steatosis, inflammation, and hepatocellular ballooning according to the Histological Scoring System for Nonalcoholic Fatty Liver Disease [19].

### 2.4. Human Study

One hundred participants affected by overweight (body mass index (BMI) z-score between the 85th and 97th percentiles (equivalent to >1 standard deviation (SD) according to WHO criteria)) or obesity (BMI z-score > 97th percentile (equivalent to >2 SD according to WHO criteria)) (47 boys and 53 girls) aged 5.7–10.2 years (mean, 8.5 years; median, 8.6 years) were voluntarily recruited between October 2021 and January 2022. At study entry, all participants underwent a physical examination, anthropometric measurements (weight, height, and waist), and standard laboratory tests including measurements of fasting glucose and insulin, total cholesterol, low-density lipoprotein-cholesterol, high-density lipoprotein-cholesterol, triglyceride, apolipoprotein, uric acid, alanine aminotransferase, and aspartate aminotransferase. Total body fat was measured by bioimpedance. Anthropometric measurements were performed again at 24 weeks (the end of the intervention period) and 12 weeks later.

Dietary advice from the dietitian, which was received by all parents and children, included the consumption of a normoenergetic diet and performance of low-level physical activity for at least 1 h daily throughout the study. Next, participants were randomly assigned to two groups. Children in the prebiotic group consumed a daily portion of fruit and vegetable mousses prepared by the Tymbark food industry (Tymbark MWS Sp. z o. o., Tymbark, Poland) enterprise supplemented with 10 g of SDexF produced from potato starch for 24 weeks, while children in the control group consumed a daily portion of these mousses lacking SDexF. Mousses were supplied to the participants in identical foil packets, and participants in both groups consumed two packets/day. The composition of the mousses and methods of SDexF supplementation are described elsewhere [20]. Randomization was performed by an investigator without involving the participants, and both participants and study staff were blinded to the treatments.

All procedures were performed following the ethical standards of the institutional and/or national research committees and conformed with the Helsinki Declaration and its later amendments or comparable ethical standards. The study was approved by the Ethics Committee of the Children’s Memorial Health Institute, Warsaw, Poland (18/KBE/2021). A complete description was provided to all participants and their parents, and written informed consent was obtained from the parents before enrollment.

The inclusion criteria were children of both sexes, aged 6–10 years; overweight or obesity according to the WHO definition; and informed consent for study participation signed by parents. The exclusion criteria were an allergy to mousse components, malabsorption syndrome, organ failure, food neophobia, and other diseases or health problems that may interfere with the study procedures or safety. None of the participants used antibiotics or probiotics within six months before and during the fecal sampling.

Fecal samples were collected from all participants at study entry, at the end of the intervention period, and 12 weeks later, i.e., at the end of the follow-up period. Stool samples were collected using a stool specimen collection kit, which consisted of a Styrofoam box containing a sterile tube with a spatula and an ice pack. The samples were transported to the laboratory at −20 °C and stored at −80 °C until use.

### 2.5. 16s-rRNA-Seq Metagenomics and Metabolomics Procedures

In both animal and human studies, metagenomic- and metabolomic-based analyses were conducted. Genomic DNA was isolated from mouse and human fecal samples using a QIAamp Fast DNA Stool Mini Kit (Qiagen, Hilden, Germany) according to the manufacturer’s protocol, as described previously [21,22]. The quality and quantity of the extracted DNA were assessed by measuring the optical density using a NanoDrop 2000/2000c spectrophotometer (Thermo Fisher Scientific, Carlsbad, CA, USA) and a fluorometric-based method using a Qubit dsDNA HS Assay Kit (Thermo Fisher Scientific), respectively. Library preparation of the variable V3 and V4 regions of the bacterial 16S rRNA gene was performed according to the 16S Metagenomic Sequencing Library Preparation protocol on an Illumina platform (Illumina, Inc., San Diego, CA, USA). Sequences were obtained on an Illumina MiSeq system in a 2 × 300 bp paired-end run.

Short-chain fatty acids (SCFAs) and amino acids (AAs) were extracted and derivatized as described previously [23]. Gas chromatographic analysis of fecal extracts was performed on an Agilent 7000D Triple Quadrupole mass spectrometer coupled with a 7890 gas chromatography (GC) system with a G4513A autosampler (Agilent Technologies, Santa Clara, CA, USA). A VF-5ms column (30 m, 0.25 mm, 0.50 µm) was used for analysis. Mass spectrometry (MS) data were gathered in full scan mode for *m*/*z* 15–650 at a frequency of 4.9 scans per second. MassHunter software version 10.0.368 (Agilent Technologies) was used for analysis.

### 2.6. Statistical Methods

DADA2 [24] pipeline version 1.26 was used for read error correction, amplicon sequence variant identification, and chimeric read identification and removal. Taxonomy assignment was conducted with Mothur version 1.43 [25] using SILVA (version 1.138 [26]). Diversity indices (Shannon and Chao1) were computed with iNEXT R package version 3.0 [27] using 1000 bootstrap samples. Significant differences between groups were identified with the Mann–Whitney U test or mixed-effects linear models (as implemented in package lmerTest version 3.1 [28]). Distances (β-diversity) were computed with vegan package version 2.6-4 [29] using the robust Aitchison metric. Significant grouping was identified with the adonis2 function. Differentially abundant amplicon sequence variants were identified with LINDA [30] (as implemented in package MicrobiomeStat version 1.1). Differences in the relative abundances of AAs and SCFAs were identified with the Multiple *t* tests (and nonparametric tests)—one per row test using GraphPad Prism v10.1.1(323).

## 3. Results

### 3.1. Animal Study

In the animal study, we determined whether SDexF supplementation may reduce the risk of weight gain in mice fed a WD that may be accompanied by changes in the composition of the gut microbiome and metabolites.

Despite a high degree of inter-individual variability in response to WD feeding, weight gain was significant in female and male mice from weeks 20 and 22, respectively, compared with control mice on an ND (Figure 2A,B). At week 28, the last time point in this experiment, liver weight was increased (Figure 2C) and liver color was visibly lighter in both WD-fed female and male mice. Hematoxylin and eosin staining showed that all ND-fed mice had normal livers, while all WD-fed mice had hepatic steatosis of different intensities. More than half of obese mice exhibited increased lobular inflammation and hepatocyte ballooning, resulting in an increased NAFLD Activity Score (NAS) [31] (Table 1).

Relative to ND-fed mice, both WD-fed male and female mice developed hypercholesterolemia (Figure 3). Mean serum aspartate transaminase, alanine transaminase, alkaline phosphatase, and glucose levels were higher in WD-fed male mice than in ND-fed male mice. Mean serum aspartate transaminase and glucose levels did not significantly differ between WD- and ND-fed female mice, while the mean serum triglyceride level was higher in WD-fed female mice than in ND-fed female mice.

Adding SDexF to the ND significantly decreased the body weight of male mice at weeks 6–9 and of female mice at week 9 and weeks 26–28 compared with ND-fed mice (Figure 4). Adding SDexF to the ND had no effect on liver weight in male or female mice (Figure 2C). In female mice, adding SDexF to the WD significantly reduced body weight gain at weeks 22–28 and liver weight at the end of the experiment. By contrast, in male mice, adding SDexF to the WD reduced body weight gain at weeks 19–22, but there were no differences in body or liver weight at the end of the experiment. The effects of SDexF on liver steatosis and serum biochemical measurements were limited and inconclusive (Table 1, Figure 2).

Next, we compared the gut microbiota and metabolites in stool samples collected from mice fed an ND and WD at the start and end of the experiment. An average of 33,489 reads were generated per sample (median, 34,836). Of the 13 identified phyla, three (Bacteroidota, Firmicutes, and Verrucomicrobiota) had an abundance higher than 1% of the microbiome. Of the 162 genera discovered, 12 had an abundance higher than 1%, and the five most prevalent were *Alistipes*, *Akkermansia*, *Muribaculaceae*, and two *Lachnospiraceae* groups.

The gut bacterial community structure of the mice was evaluated by analyzing α- and β-diversity of the fecal microbiota. α-diversity was analyzed using the Shannon index, a marker of bacterial richness and evenness, and the Chao1 index, a marker of bacterial richness. β-diversity was analyzed using the principal co-ordinate analysis (PCoA) of Bray–Curtis distances. The analyses were performed at the genus level.

After multiple hypothesis testing corrections, the Shannon index did not differ between the beginning and end of the experiment in ND-fed female mice, but was lower at the end of the experiment than at the beginning of the experiment in ND-fed male mice (Figure 5A,C). A significant difference in the Chao1 index between the beginning and end of the experiment was observed in ND-fed female, but not in male, mice (Figure 5B,D). Twenty-eight weeks of feeding a WD significantly increased the Shannon index in female mice but not in male mice (Figure 5E,G), and enhanced the Chao1 index in both female and male mice (Figure 5F,H). β-diversity of the bacterial structure formed separate clusters, with significantly different microbial compositions at the end of the experiment compared with the beginning of the experiment in mice of both sexes regardless of diet (Figure 6).

At the family level, the abundances of nine (four overrepresented, five underrepresented), six (three overrepresented, three underrepresented), twenty-one (seven overrepresented, fourteen underrepresented), and nineteen (seven overrepresented, twelve underrepresented) taxa significantly differed between the beginning and end of the experiment in ND-fed female, ND-fed male, WD-fed female, and WD-fed male mice, respectively (adjusted *p* < 0.05, Appendix A). Of these, *Lactobacillaceae* was underrepresented in ND-fed mice of both sexes, while *Butyricicoccaceae*, *Prevotellaceae*, *Helicobacteraceae*, *Atopobiaceae*, and *Sutterellaceae* were underrepresented and *Peptostreptococcaceae* and *Christensenellaceae* were overrepresented in WD-fed mice of both sexes. A similar comparison performed at the genus level uncovered twenty-one (six overrepresented, fifteen underrepresented), three (one overrepresented, two underrepresented), thirty-five (eleven overrepresented, twenty-four underrepresented), and thirty-eight (eighteen overrepresented, twenty underrepresented) taxa in ND-fed female, ND-fed male, WD-fed female, and WD-fed male mice, respectively (adjusted *p* < 0.05, Appendix A). Of these, *Candidatus_Stoquefichus* and *Roseburia* were underrepresented in ND-fed mice of both sexes, while the abundances of 16 taxa (*Helicobacter*, *Ruminococcus*, *Parasutterella*, *Turicibacter*, *Limosilactobacillus*, *Lachnospiraceae_UCG-001*, *Lachnospiraceae_UCG-006*, *Muribaculum*, *Lactobacillus*, *Romboutsia*, *Bilophila*, *Prevotellaceae_UCG-001*, *UBA1819*, *Butyricicoccus*, *Colidextribacter*, and *Alloprevotella*) were changed in WD-fed mice of both sexes.

GC/MS-based analyses identified seven SCFAs (formic acid, acetic acid, propanoic acid, isobutyric acid, butanoic acid, pentanoic acid, and hexanoic acid) and ten AAs (alanine, glycine, glutamic acid, isoleucine, leucine, methionine, phenylalanine, proline, valine, and tyrosine) in extracts from fecal samples.

Compared to the relative fecal abundances observed in mice fed an ND, feeding with a WD for 28 weeks resulted in a significant increase in the abundances of three SCFAs (acetic acid, propanoic acid, and butyric acid) and three AAs (valine, leucine, and isoleucine) in both females and males, and increased abundances of three other SCFAs (formic acid, isobutyric acid, and hexanoic acids) and four AAs (glycine, methionine, phenylalanine, and tyrosine) found only in males (Figure 7).

The effect of SDexF on the gut bacterial community was analyzed by comparing the microbiota structure between SDexF-treated and untreated mice at the end of the 28-week experiment. SDexF significantly changed bacterial β-diversity in mice of both sexes fed an ND and WD (Figure 8), and decreased the Shannon index in both groups of females supplemented with the SDexF. (Figure 9A,E). In ND-fed males SDexF did not affect the Shannon index but in WD-fed males it did (Figure 9C,G). SDexF supplementation decreased the Chao index in WD-fed mice of both sexes (Figure 9F,H), but not in the ND-fed groups (Figure 9B,D). In addition, bacterial abundance differed between SDexF-treated and untreated mice. The abundances of nine, one, four, and four taxa at the family level and six, zero, six, and fifteen taxa at the genus level differed between SDexF-treated and untreated ND-fed female, ND-fed male, WD-fed female, and WD-fed male mice, respectively (Appendix A)

The specific effects of SDexF on taxa abundances were finally determined by comparing the lists of bacteria that differentiated the gut microbiomes of mice at the beginning and end of the experiment with those that differentiated the gut microbiomes of SDexF-treated and untreated mice. SDexF specifically affected five (*Enterobacteriaceae*, *Muribaculaceae*, *Rikenellaceae*, *Bacteroidaceae*, and *Tannerellaceae*), two (*Tannerellaceae* and *Clostridia_UCG-014_fa_*), and two (*Erysipelotrichaceae* and *Tannerellaceae*) family-level bacteria found in ND-fed female, WD-fed female, and WD-fed male mice, respectively, and two (*Anaerotruncus* and *Tannerellaceae_unclassified*), two (*Paenalcaligenes* and *Parabacteroides*), and five (*Prevotella_9*, *Dialister*, *Family_XIII_AD3011_group*, *Dubosiella*, and *Gemella*) genus-level bacteria found in ND-fed female, WD-fed female, and WD-fed male mice, respectively (Appendix A).

Supplementation with SDexF decreased the relative fecal abundances of pentanoic acid and all AAs in ND-fed female mice, while increasing propanoic acid and decreasing the abundances of alanine, valine, leucine, isoleucine, and glutamic acid in WD-fed females. In male mice fed an ND, SDexF increased the abundance of acetic acid, and in males fed a WD, increased the abundancies of acetic, propanoic, and butyric acids, as presented in the Appendix A.

### 3.2. Human Study

In the human study, we determined whether SDexF supplementation can support weight gain and changes in intestinal microbiota composition in children with overweight or obesity during a weight loss intervention that introduced dietary restrictions and properly dosed physical exercise. This research was carried out on the occasion of a larger randomized, double-blind clinical trial (registered on ClinicalTrials.gov, Identifier: NCT05140070), for which detailed clinical results will be reported in a separate publication. In this study, we focused on the presentation of fecal metagenomic and metabolomic data in relation to SDexF supplementation.

Of 100 participants, 39 in the prebiotic group and 42 in the control group provided stool samples. At baseline, sex, height, weight, and BMI z-score did not significantly differ between the two groups (Table 2). At the start of the study, in children randomized to the SDexF and placebo group, the percentage of body fat was 26.4% and 26.6%, respectively, while the percentage of muscle mass was 40.6% and 40.5%, respectively. At the end of treatment, in children treated with dextrin and placebo, the percentage of body fat was 26.4% and 26.1%, respectively, while the percentage of muscle mass was 40.6% and 40.8%.

Patients were assessed every 3 months by anthropometric measurements, and standardized nutritional and physical activity guidance was provided by a research team (physician, psychologist, and dietician). Although absolute body weight increased in all participants over 24 weeks of SDexF supplementation and continued for another 12 weeks after the supplementation ended, weight gain was slowed in a similar percentage of children in both groups at these two time points compared with baseline (Table 3). Thus, while lifestyle modification is sufficient to reduce weight gain in more than half of overweight/obese children, SDexF has no effect on weight gain.

DNA and metabolites isolated from fecal samples were analyzed using 16s-rRNA sequencing and GC/MS analysis. An average of 40.5 million reads were generated per sample (median, 40 million). Four (Bacteroidota, Firmicutes, Proteobacteria, and Actinobacteriota) of the eleven identified phyla had an abundance higher than 1% of the microbiome. Of the 163 genera discovered, 14 had a mean abundance higher than 1%.

After multiple hypothesis testing corrections, microbiome measures, including α- and α-diversity and the abundances of individual bacteria, did not differ between the two groups of participants at baseline (Figure 10). In addition, both α- and β-diversity was unaffected by SDexF supplementation and BMI z-score changes in all of the comparisons with one exception (Figure 10). The Shannon index showed that α-diversity significantly differed (adjusted *p* = 0.034, fold change = 1.08) between control children with and without slowed weight gain at week 24 (Table 4).

The abundances of only single bacteria differentiated the analyzed groups. At the genus level, *Escherichia-Shigella* was (adjusted *p* < 0.05), or tended to be (adjusted *p* < 0.1), overrepresented at weeks 24 and 36 in the prebiotic group and at week 36 in the control group, while *Erysipelotrichaceae UCG-003* was underrepresented at week 24 in the control group compared with baseline. At week 24, the abundance of *Streptococcus* differed between children in the control group with and without slowed weight gain and between children in the prebiotic group and children in the control group without slowed weight gain. *Lachnoclostridium*, *Butyricicoccus*, and *Erysipelotrichaceae* were, or tended to be, underrepresented in children in the prebiotic group compared with children in the control group, both with slowed weight gain at week 24. At week 36, *Lachnoclostridium* was significantly overrepresented in children in the prebiotic group with slowed weight gain compared with those without slowed weight gain (Table 5).

As in the mouse studies, GC/MS-based analysis of fecal samples identified seven SCFAs and ten AAs. However, SDexF affected neither the levels of SCFAs nor the levels of AAs in any of the pair-wise comparisons (not shown). Instead, the daily consumption of vegetable and fruit mousses, regardless of whether they were supplemented with SDexF (Figure 11, down panel) or not (Figure 11, up panel), significantly lowered relative AA levels at week 24. This effect was partially maintained at week 12 after the end of the study in both prebiotic and control groups.

## 4. Discussion

While an obesogenic WD plays a critical role in remodeling the gut microbiome, foods rich in prebiotics may stimulate the intestinal growth of Lactobacillus and Bifidobacterium and reduce bacteria belonging to the phyla Firmicutes and Bacteroidetes that can prevent or even treat overweight and obesity [8,32]. SDexF is a short-chain glucose polymer obtained by the modification of potato starch [20]. Similarly to other dietary fibers, SDexF exhibits increased resistance to hydrolysis by mammalian digestive enzymes and, once fermented by colonic microbiota, can stimulate bacterial growth and/or beneficial responses to the host [13,33]. As reported previously, the relative abundance of *Tenericutes* was lower and the abundances of *Verrucomicrobia*, *Verrucomicrobiaceae*, and *Akkermansia* were significantly higher in dextrin-treated mice than in non-dextrin-treated mice fed a high-fat, high-fructose (HFHF) diet [34]. Treatment with resistant dextrin significantly relieved liver insulin resistance, improved serum lipid profiles, and reduced liver triglyceride and total cholesterol levels. Using a colitis model, Valcheva et al. showed that the abundances of butyrate producers, *Lachnospiraceae* and *Ruminococcaceae*, were increased in mice fed a dextrin-based diet [35]. These findings confirm that a diet rich in soluble prebiotic fibers can stimulate the growth of specific components of the gut microbiota and help reduce the levels of inflammatory cytokines even in obese individuals. Thus, starch degradation products are highlighted as prebiotics [13].

This study was conducted to assess whether SDexF (soluble dextrin fiber) can support weight loss in overweight/obese children who underwent an obesity intervention and whether it may reduce the risk of weight gain in mice fed a WD. Furthermore, we determined whether these possible effects are accompanied by changes in the composition of the gut microbiome and metabolites.

We first confirmed that the feeding of WD induced body weight gain and liver steatosis in mice of both sexes. WD feeding modulated both α- and β-diversity of the fecal microbiota. The Shannon index was significantly increased in obese female mice, and the Chao index was enhanced in obese mice of both sexes, while β-diversity of microbial compositions differed between the end and beginning of the experiment in mice of both sexes regardless of diet. Among numerous taxa whose abundances differed significantly between the beginning and end of the experiment, a WD decreased the abundances of *Butyricicoccaceae*, *Prevotellaceae*, *Helicobacteraceae*, *Atopobiaceae*, and *Sutterellaceae* and increased the abundances of *Peptostreptococcaceae* and *Christensenellaceae* at the family level. At the genus level, the abundances of 16 taxa (*Helicobacter*, *Ruminococcus*, *Parasutterella*, *Turicibacter*, *Limosilactobacillus*, *Lachnospiraceae_UCG-001*, *Lachnospiraceae_UCG-006*, *Muribaculum*, *Ligilactobacillus*, *Romboutsia*, *Bilophila*, *Prevotellaceae_UCG-001*, *UBA1819*, *Butyricicoccus*, *Colidextribacter*, and *Alloprevotella*) were changed in mice of both sexes.

The gut microbiota is implicated in the control of food intake and satiety [36], and studies in mice have shown that the obesity-related composition of the intestinal microbiota, with increased abundances of *Bacteroides*, *Roseburia*, *Bifidobacterium*, *Fecalibacterium*, and *Enterobacteria*, improves the ability to absorb energy from the diet and provide energy and nutrients for microbial growth and proliferation [37,38,39]. They typically ferment undigested carbohydrates and synthesize SCFAs, including butyrate, propionate, and acetate [37]. Acetate enters the systemic circulation and reaches peripheral tissues, while propionate and butyrate are used mainly in the liver and intestinal epithelium, respectively, as an energy source [13]. Transplantation of the gut microbiome from obese mice, for genetic or diet-related reasons, into normal weight mice increased their fat mass and consequently body weight [40]. When transplanted with the gut microbiome of an individual with or without obesity, mice lacking the gut microbiota acquired a weight-related phenotype from the donor [41].

Next, we confirmed that prolonged SDexF treatment slowed body weight gain in female mice fed an ND or WD, while reduction in weight gain was transient and not observed at the end of the experiment in male mice fed an ND and WD. The liver weight of WD-fed female, but not male, mice was decreased compared with controls. SDexF significantly changed bacterial α-diversity in mice of both sexes fed an ND or WD and decreased the Shannon index in females fed an ND and WD supplemented with SDexF. Furthermore, SDexF supplementation affected the Shannon index in males fed a WD. The Chao index decreased in WD-fed mice of both sexes. Furthermore, SDexF specifically affected the fecal abundances of five (*Enterobacteriaceae*, *Muribaculaceae*, *Rikenellaceae*, *Bacteroidaceae*, and *Tannerellaceae*), two (*Tannerellaceae* and *Clostridia_UCG-014_fa_*), and two (*Erysipelotrichaceae* and *Tannerellaceae*) taxa at the family level in ND-fed female, WD-fed female, and WD-fed male mice, respectively, and two (*Anaerotruncus* and *Tannerellaceae_unclassified*), two (*Paenalcaligenes* and *Parabacteroides*), and five (*Prevotella_9*, *Dialister*, *Family_XIII_AD3011_group*, *Dubosiella,* and *Gemella*) taxa at the genus level in ND-fed female, WD-fed female, and WD-fed male mice, respectively. Finally, SDexF decreased the relative fecal abundances of pentanoic acid and all AAs and increased the abundance of acetic acid in ND-fed female and male mice, respectively, and increased propanoic acid and decreased the abundances of alanine, valine, leucine, isoleucine, and glutamic acid in females fed a WD, while increasing the abundances of acetic acid, propanoic acid, and butyric acid in males fed a WD.

Gonadal hormones can alter the gut microbiome that has been reported in humans and rodents [42]. Significant sex differences in lipid metabolism and microbiota composition at baseline, along with sex-dependent responses to a high-fat diet (HFD), have recently been reported [42]. Female mice exhibited less body weight gain and body fat composition, accompanied by an enriched growth of beneficial microbes (e.g., *Akkermansia*) and depleted growth of *Adlercreutzia* and *Enterococcus*. As previously reported, sex can have a greater impact on the composition of the gut microbiota composition than HFD and antibiotics [43]. Furthermore, prebiotics have been shown to alleviate anxiety in female mice by changing the composition of the gut microbiota composition in a sex-specific manner [44], and the sexual dimorphism of the axis of *gut microbiota–host metabolism* mainly involves changes in the metabolism of SCFAs and amino acids [45]. However, the final mechanisms by which the gut microbiota can associate with a sexually dimorphic response to an HFD are still unclear.

In contrast to the pronounced impacts of SDexF on weight gain and gut microbiome and metabolome composition found in the animal study, which were observed particularly in female mice, SDexF did not affect weight loss or gut metagenomic and metabolomic profiles in children, affecting only the abundances of individual taxa. Instead, the daily consumption of vegetable and fruit mousses significantly reduced relative AA levels at week 24 of the study regardless of SDexF treatment. This effect was partially maintained at week 12 after the end of the study in both the prebiotic and control groups. However, the clinical significance of this observation is unclear.

SDexF, characterized by its soluble fiber-like qualities, remains indigestible to human digestive enzymes as a result of the glycosidic bonds present in its structure, which avoid breakdown by conventional amylolytic enzymes. Therefore, we expected that these substances would reach the large intestine unchanged. Since SDexF has additional documented prebiotic properties, it should promote the production of short-chain fatty acids. Unfortunately, this effect was observed only in animal studies and was not confirmed in studies on human feces.

The efficacy of fiber supplements for the management of obesity management is dependent on the content and physicochemical properties of the fibers, such as their solubility, fermentability, and viscosity [46]. Clinical trials showed that two of the most popular commercially available resistant dextrins, Nutriose, a soluble fiber made from wheat, maize, or pea starch, and Fibersol-2, a soluble maltodextrin produced from corn-originated starch, have multiple positive effects on various health markers [16]. Both prebiotics have beneficial effects on BMI and weight loss in overweight adults, improve glucose and lipid metabolism, stimulate the production of satiety peptides, improve serum immunologic indicators, and reduce endotoxin and inflammation levels. They also have beneficial effects on the gut microbiota, such as promoting the growth of health-beneficial bacteria and increasing SCFA production [16].

Most studies investigating the effects of prebiotics on obesity prevention were performed in animal models of obesity, while those investigating the effects of prebiotics on obesity treatment were performed in overweight/obese humans. A similar strategy was adopted in our study. Research in animal models of obesity allows the use of procedures that are considered ethically unacceptable in human research, such as fattening animals using a diet that is generally considered harmful and supplementing animals with the prebiotic tested at doses that are unacceptable in humans. In our studies, we fed mice a diet supplemented with SDexF at a level of 5 g of SDexF per 100 g of feed, while the acceptable daily dose of SDexF in children was established at 10 g/two packets of mousse. Thus, the daily dose of SDexF consumed by children was significantly lower than that consumed by mice.

Although it is unclear whether the dose of 10 g of SDexF a day was insufficient to induce weight loss effects in our participants, significant differences in the effects of SDexF observed in mice and children could be due to a higher SDexF content in the diet of mice than in children. It also cannot be ruled out that SDexF administered to mice in higher concentrations resulted in lower consumption of the diet containing SDexF than that without SDexF. Unfortunately, we had not consistently monitored dietary intake between SDexF-consuming groups and those who did not. However, sporadic evaluations of dietary intake during the 28-week experimental period did not show significant differences.

As reported, fiber supplements without weight loss interventions did not cause considerable anti-obesity effects [47]. In our human study, SDexF supplementation was an addition to the weight loss intervention that primarily involved the modification of lifestyle to change the physical activity and eating behaviors of the participants by intensive training under the supervision of a research team. As a consequence, it cannot be excluded that the potential effect of SDexF was masked by other elements of the intervention program or that the dose of SDexF, which was well tolerated by the participants, may not have been sufficient for the weight loss effect. However, our research protocol included a qualitative rather than quantitative measurement of food intake by children exposed to SDexF which may be considered a limitation of our work. In total, 60% of the participants achieved weight loss regardless of SDexF treatment at the end of the intervention and 55% of the participants maintained weight loss for 12 weeks after the intervention.

## 5. Conclusions

Although mouse models of obesity are commonly accepted in studies on diet and other environmental factors, there are large differences in the contribution of the intestinal microbiota to metabolic rate and dietary habits between humans and mice. However, such models are utilized due to the absence of superior alternatives and, therefore, the results from the mice studies should be extrapolated to humans with caution [48]. While the results of our study may come as a surprise, they do not definitively negate the potential of SDexF to aid in weight loss among children. More research is warranted to conclusively establish whether SDexF supplementation is beneficial in obesity-targeting interventions.

## Figures and Tables

**Figure 1 nutrients-16-00917-f001:**
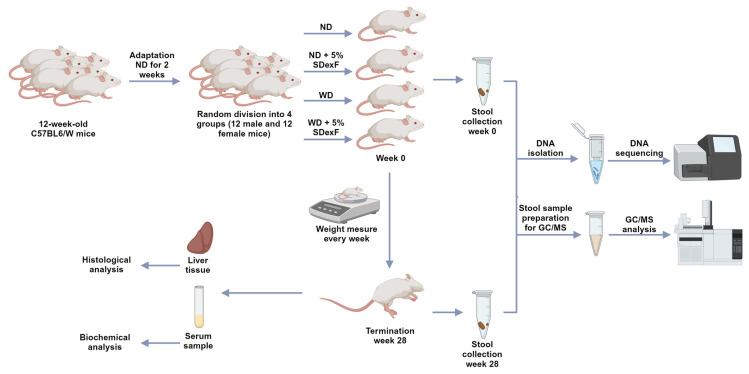
Flow chart of mice experiment (created with BioRender.com, accessed on 4 March 2024).

**Figure 2 nutrients-16-00917-f002:**
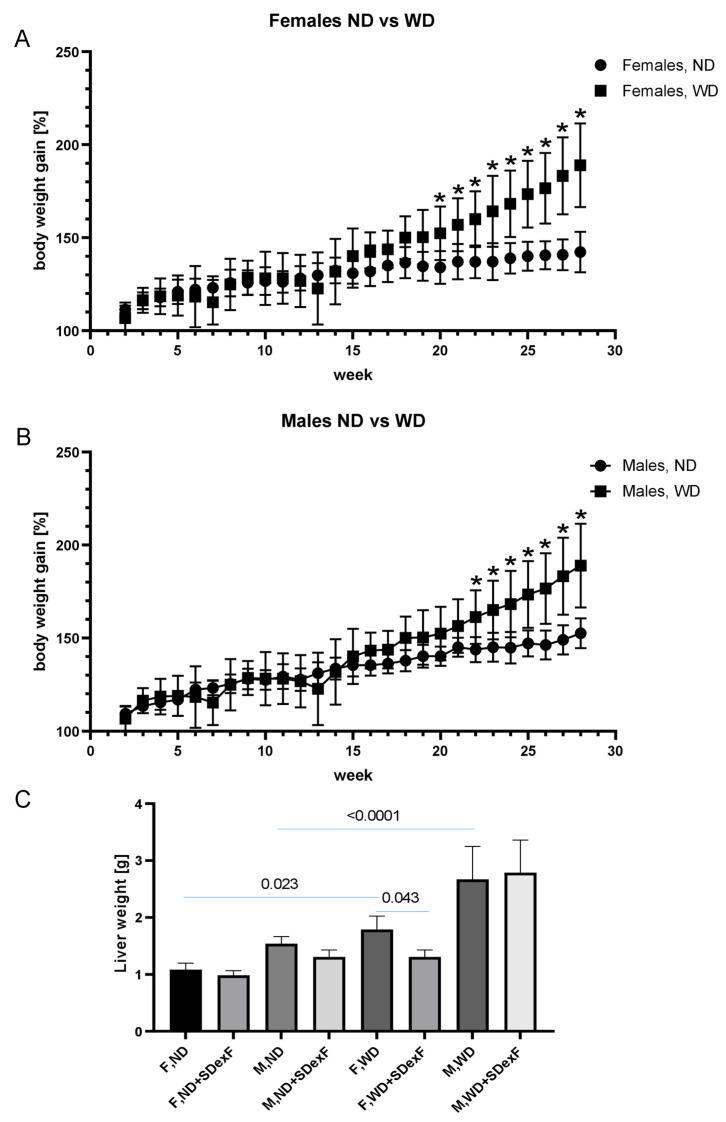
Changes in body weight of Normal Diet (ND) (**A**) and Western diet (WD) (**B**) fed mice. Liver weights in the studied mouse groups at the end of the 28-week experiment (**C**). F, Female mice; M, Male mice; SDexF, soluble dextrin fiber. * *p* < 0.05.

**Figure 3 nutrients-16-00917-f003:**
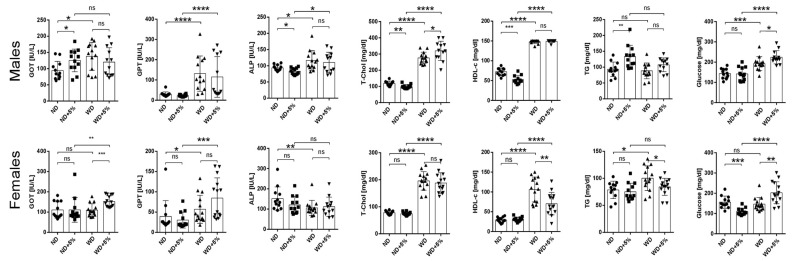
Serum biochemical markers in mice measured at the end of the 28-week experiment. GOT, glutamic oxaloacetic transaminase; GPT, glutamic pyruvate transaminase; ALP, alkaline phosphatase; T-Chol, total cholesterol; HDL, high density lipoprotein; TG, triglyceride. * *p* < 0.05; ** *p* < 0.01; *** *p* < 0.001; **** *p* < 0.0001; ns, not significant.

**Figure 4 nutrients-16-00917-f004:**
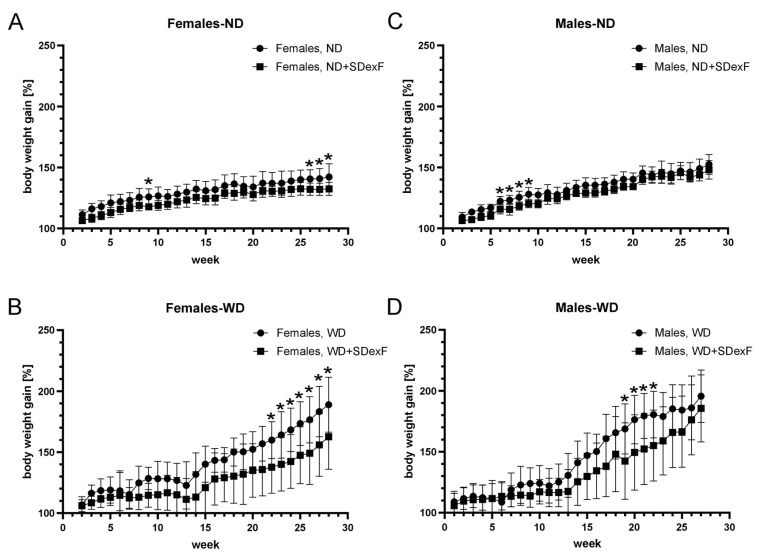
Effects of soluble dextrin fiber (SDexF) on body weight gain in ND-fed female (**A**) and male (**C**) and WD-fed female (**B**) and male (**D**) mice. * *p* < 0.05.

**Figure 5 nutrients-16-00917-f005:**
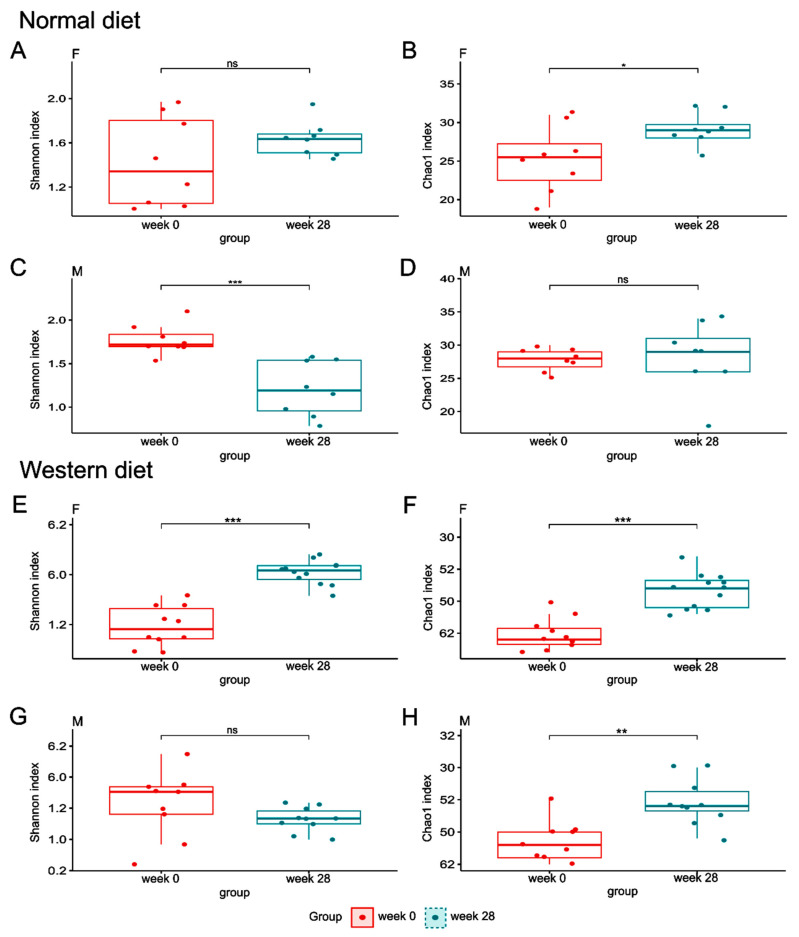
α-diversity measured by the Shannon and Chao1 indexes in ND-fed female (**A**,**B**), ND-fed male (**C**,**D**), WD-fed female (**E**,**F**), and WD-fed male (**G**,**H**) mice. * *p* < 0.05; ** *p* < 0.01; *** *p* < 0.001; ns, not significant.

**Figure 6 nutrients-16-00917-f006:**
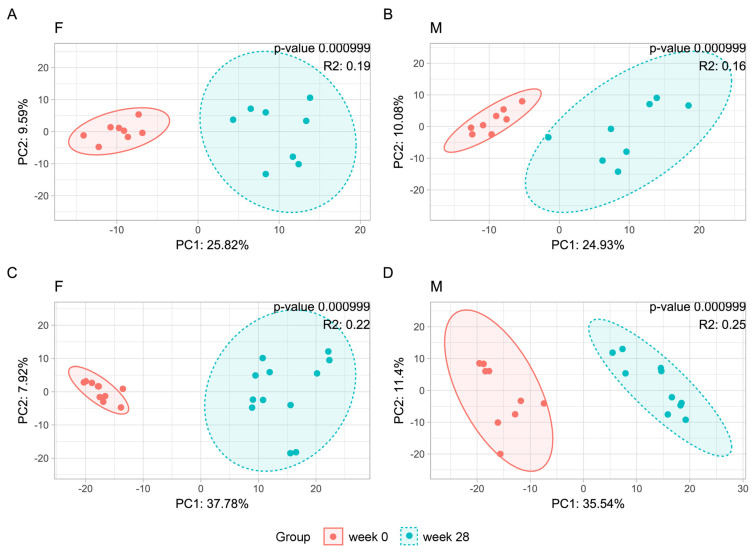
β-diversity measured by principal co-ordinate analysis (PCoA) in ND– and WD-fed female (**A**,**C**) and male (**B**,**D**) mice.

**Figure 7 nutrients-16-00917-f007:**
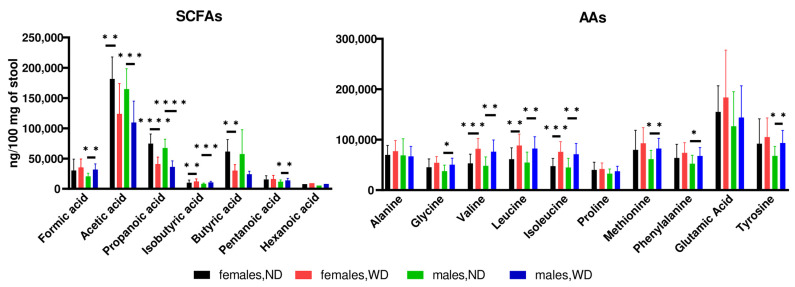
The relative fecal abundances of short-chain fatty acids (SCFAs) and amino acids (AAs) at the end of the experiment in female and male mice fed a normal diet (ND) and Western diet (WD), presented as mean and SD. * *p* < 0.05; ** *p* < 0.01; *** *p* < 0.001; **** *p* < 0.0001.

**Figure 8 nutrients-16-00917-f008:**
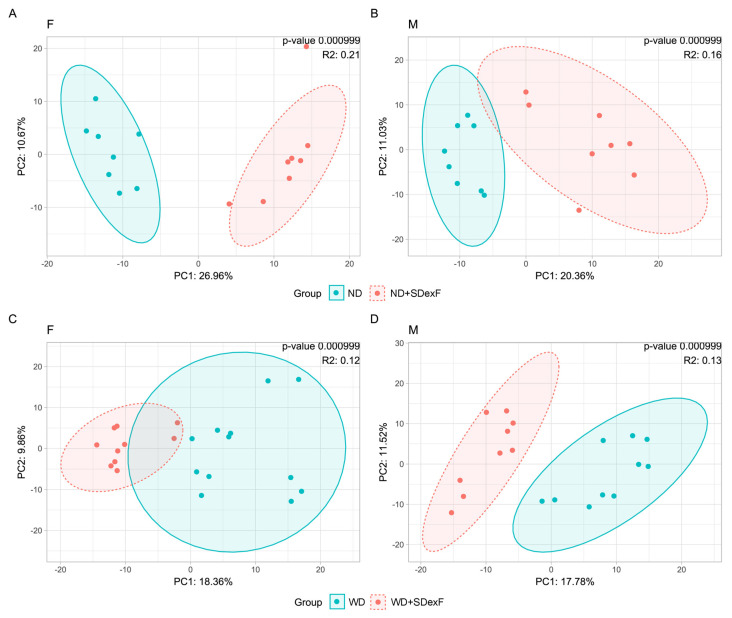
Effects of SDexF on β-diversity measured by PCoA in ND– and WD–fed female (**A**,**C**) and male (**B**,**D**) mice.

**Figure 9 nutrients-16-00917-f009:**
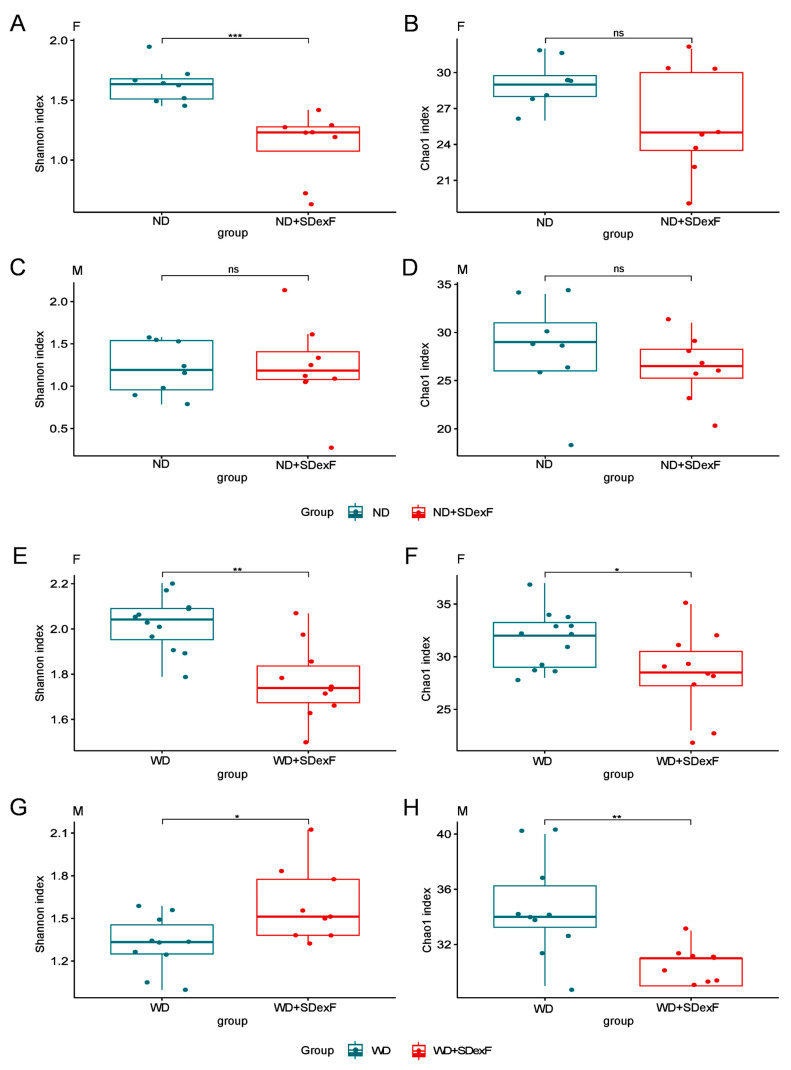
Effects of SDexF on α-diversity measured by the Shannon and Chao1 indexes in ND-fed female (**A**,**B**), ND-fed male (**C**,**D**), WD-fed female (**E**,**F**), and WD-fed male (**G**,**H**) mice. * *p* < 0.05; ** *p* < 0.01; *** *p* < 0.001; ns, not significant.

**Figure 10 nutrients-16-00917-f010:**
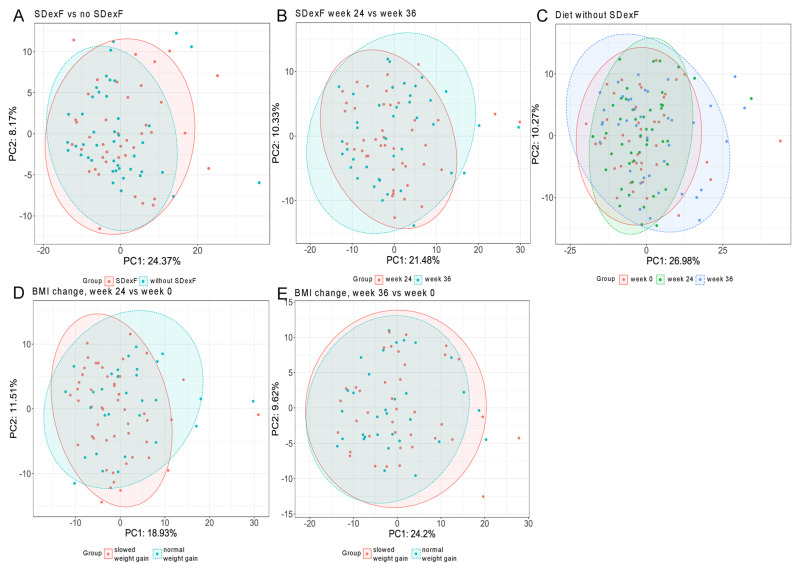
PCoA of Bray-Curtis distances. Samples from (**A**) the SDexF and control groups at baseline; (**B**) the SDexF group at weeks 24 and 36; (**C**) the control group at weeks 0, 24, and 36; and the SDexF-treated and control children with and without slowed weight gain at week 24 (**D**) and week 36 (**E**).

**Figure 11 nutrients-16-00917-f011:**
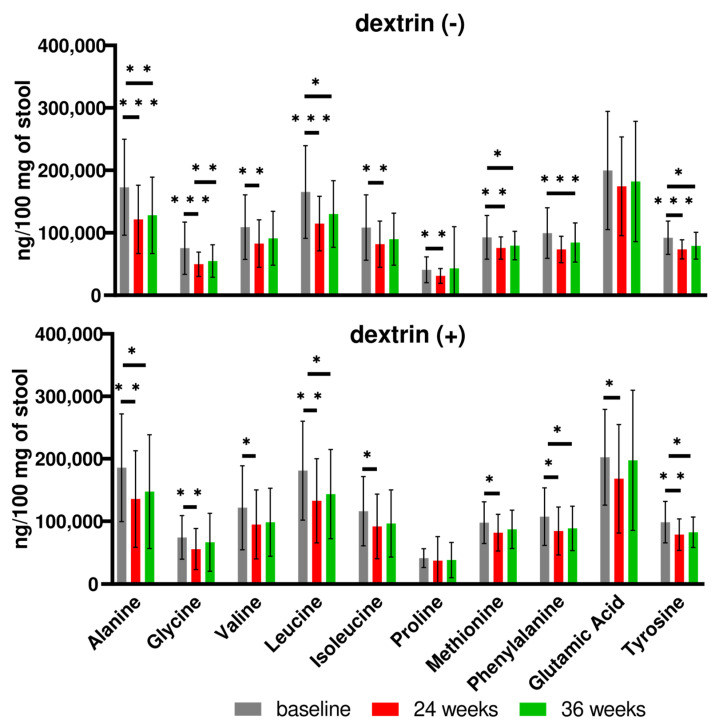
Effect of consumption of vegetable and fruit mousses for 24 weeks on the relative fecal abundances of amino acids (AAs) in children not supplemented (**up panel**) or supplemented with SDexF (**down panel**), presented as mean and SD. * *p* < 0.05; ** *p* < 0.01; *** *p* < 0.001.

**Table 1 nutrients-16-00917-t001:** Effects of feeding a normal diet (ND) and Western diet (WD) supplemented with or without soluble dextrin fiber (SDexF) for 28 weeks on the intensities of liver steatosis, hepatic lobular inflammation, and liver cell ballooning injury in mice according to the Non-alcoholic fatty liver disease (NAFLD) Activity Score.

Feeding	Sex	Score	Steatosis				Lobular Inflammation				Hepatocellular Ballooning		
			0	1	2	3	0	1	2	3	0	1	2
ND	F	N=	12	0	0	0	10	2	0	0	12	0	0
	M	N=	12	0	0	0	9	3	0	0	12	0	0
ND + SDexF	F	N=	12	0	0	0	7	5	0	0	12	0	0
	M	N=	12	0	0	0	8	4	0	0	12	0	0
WD	F	N=	0	1	6	5	1	3	7	1	6	6	0
	M	N=	0	0	2	10	0	3	8	0	5	7	0
WD + SDexF	F	N=	0	5	4	3	3	4	6	0	2	10	0
	M	N=	0	3	2	7	5	7	0	0	7	5	0

F, female; M, male; Steatosis scoring system: 0, <5%; 1, 5–33%; 2, >33–66%; 3, >66%; Lobular inflammation scoring system: 0, no foci; 1, <2 foci per 200× field; 2, 2–4 foci per 200× field; 3, >4 foci per 200× field; Hepatocellular ballooning scoring system: 0, none; 1, few balloon cells; 2, many cells/prominent ballooning.

**Table 2 nutrients-16-00917-t002:** Baseline demographic characteristics of children participants in the two groups.

	Prebiotic Group (*n* = 39)	Control Group (*n* = 42)	*p* Value
Age (years), mean (SD)	8.6 (1.19)	8.3 (1.18)	0.13 (Mann–Whitney U test)
Height (cm), mean (SD)	141.4 (10.8)	137.6 (9.3)	0.1 (Student’s *t* test)
Weight (kg), mean (SD)	49.8 (11.6)	46.5 (11.8)	0.15 (Mann–Whitney U test)
BMI (kg/m^2^), mean (SD)	24.6 (3.4)	24.3 (4.1)	0.43 (Mann–Whitney U test)
BMI z-score, mean (SD)	3.0 (0.8)	3.1 (1.5)	0.47 (Mann–Whitney U test)
Boys, *n* (%)	23 (54.8%)	19 (45.2%)	0.22 (Chi-squared test)
Girls, *n* (%)	16 (41.0%)	23 (59.0%)	0.22 (Chi-squared test)

BMI, body mass index; SD, standard deviation.

**Table 3 nutrients-16-00917-t003:** Comparison of the numbers of children participants with slowed body weight gain (body mass index (BMI)-for-age z-score ≤ −5% of baseline) and those whose body weight did not slow (BMI-for-age z-score > −5% and <5% of baseline) or increased (BMI-for-age z-score ≥ 5% of baseline) after 6 and 9 months between the prebiotic and control groups.

Changes in Body Weight Gain	Prebiotic Group		Control Group	
	After 6 Months N = 39	After 9 Months N = 36	After 6 Months N = 42	After 9 Months N = 40
Slowed, *n* (%)	25 (64.1%)	20 (55.6%)	25 (59.5%)	22 (55%)
Unchanged or gained, *n* (%)	14 (35.9%)	16 (44.4%)	17 (30.5%)	18 (45%)

**Table 4 nutrients-16-00917-t004:** Comparison of α-diversity in children according to soluble dextrin fiber (SDexF) supplementation and changes in body weight.

Pair-Wise Comparison	Shannon Index	Chao Index
	Adjusted *p* Value (Mean Values)	Adjusted *p* Value (Mean Values)
SDexF-supplemented group vs. control group at baseline	0.76	0.91
Comparison of stool samples collected at week 24		
SDexF-supplemented group, sample 2 vs. baseline	0.70	0.44
Control group, sample 2 vs. baseline	0.31	0.46
SDexF-supplemented group, with vs. without slowed weight gain at week 24	0.96 (2.39 vs. 2.35)	0.98 (55.7 vs. 54.1)
Control group, with vs. without slowed weight gain at week 24	0.034 (2.52 vs. 2.34)	0.11 (57.3 vs. 53.0)
SDexF-supplemented group vs. control group, both with slowed weight gain at week 24	0.081 (2.39 vs. 2.52)	0.42 (55.7 vs. 57.3)
SDexF-supplemented group vs. control group, both without slowed weight gain at week 24	0.71 (2.35 vs. 2.34)	0.46 (54.1 vs. 53)
Comparison of stool samples collected at week 36		
SDexF-supplemented group, sample 3 vs. baseline	0.84	0.22
Control group, sample 3 vs. baseline	0.49	0.53
SDexF-supplemented group, with vs. without slowed weight gain at week 36	0.77 (2.39 vs. 2.38)	0.67 (56.8 vs. 57.8)
Control group, with vs. without slowed weight gain at week 36	0.25 (2.51 vs. 2.42)	0.73 (56.2 vs. 57.8)
SDexF-supplemented group vs. control group, both with slowed weight gain at week 36	0.39 (2.39 vs. 2.51)	0.72 (56.8 vs. 56.2)
SDexF-supplemented group vs. control group, both without slowed weight gain at week 36	0.56 (2.38 vs. 2.42)	0.79 (55.3 vs. 57.8)

Sample 2, fecal sample collected at week 24; sample 3, fecal sample collected at week 36.

**Table 5 nutrients-16-00917-t005:** Comparison of bacterial abundance in children according to soluble dextrin fiber (SDexF) supplementation and changes in body weight.

Pair-Wise Comparison	Genus (Phylum)	Fold Change; Adjusted *p* Value (Mean Normalized Counts Group 1, Group 2)
SDexF-supplemented group vs. control group at baseline	None	(-)
Comparison of stool samples collected at week 24		
SDexF-supplemented group, sample 2 vs. baseline	Genus, *Escherichia-Shigella* (phylum, Proteobacteria)	3.53; 0.0701 (162, 42.3)
Control group, sample 2 vs. baseline	Genus, *Erysipelotrichaceae UCG-003* (phylum, Firmicutes)	0.37; 0.0036 (36.4, 101)
SDexF-supplemented group, with vs. without slowed weight gain at week 24	None	(-)
Control group, with vs. without slowed weight gain at week 24	Genus, *Streptococcus* (phylum, Firmicutes)	7.5; 0.0013 (17.5, 2.0)
SDexF-supplemented group vs. control group, both with slowed weight gain at week 24	Genus, *Lachnoclostridium* (phylum, Firmicutes) Genus, *Butyricicoccus* (phylum, Firmicutes) Genus, *Erysipelotrichaceae UCG-003* (phylum, Firmicutes)	3.52; 0.0211 (57.4, 15.6) 2.58; 0.0591 (62.4, 23.7) 2.36; 0.0853 (82.5, 34.3)
SDexF-supplemented group vs. control group, both without slowed weight gain at week 24	Genus, *Streptococcus* (phylum, Firmicutes)	7.99; 0.0062 (19.0, 2.0)
Comparison of stool samples collected at week 36		
SDexF-supplemented group, sample 3 vs. baseline	Genus, *Escherichia-Shigella* (phylum, Proteobacteria)	3.61; 0.0679 (167, 42.3)
Control group, sample 3 vs. baseline	Genus, *Escherichia-Shigella* (phylum, Proteobacteria)	4.59; 0.0039 (207, 62.4)
SDexF-supplemented group, with vs. without slowed weight gain at week 36	Genus, *Lachnoclostridium* (phylum Firmicutes)	0.71; 0.0378 (67.2, 95.6)
Control group, with vs. without slowed weight gain at week 36	None	(-)
SDexF-supplemented group vs. control group, both with slowed weight gain at week 36	None	(-)
SDexF-supplemented group vs. control group, both without slowed weight gain at week 36	None	(-)

Sample 2, fecal sample collected at week 24; sample 3, fecal sample collected at week 36. Slowed body weight gain, body mass index (BMI)-for-age z-score ≤ −5% of baseline; without slowed body weight gain, BMI-for-age z-score > −5% and <5% of baseline or BMI-for-age z-score ≥ 5% of baseline.

## Data Availability

The datasets presented in this study can be found in online repositories. The names of the repository/repositories and accession number(s) can be found below: https://www.ncbi.nlm.nih.gov/PRJNA1020092 and https://www.ncbi.nlm.nih.gov/PRJNA1025034, accessed on 10 February 2024.

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
