# Peer review of "Effects of Soluble Dextrin Fiber from Potato Starch on Body Weight and Associated Gut Dysbiosis Are Evident in Western Diet-Fed Mice but Not in Overweight/Obese Children"

_nutrients, 2024, doi:10.3390/nu16070917_

Round 1
Reviewer 1 Report
Comments and Suggestions for Authors
To the authors
This manuscript is reporting the effects of dietary fiber from potato starch on mice and human children. The results were different between mice and children. This manuscript may contribute for the readers to understand the role of prebiotics in obesity. There are some comments.
1. The authors showed body weight change during mouse study. I wonder if SDexF supplementation changed the tase of animal foods. The authors should show foof intake of eight groups.
2. The authors showed the result of liver fat with histological assessment. I think that measurement of liver TG content is more objective.
3. In my case, Fig. 2 does not fit on the page.
4. Tables shows p-values. However, p-value is not sufficient to understand research results (American Statistical Association). The authors should show their results as raw data, such as values of index and microbial abundance.
5. Discussion is redundant and vague. Please reconstruct it.
6. The authors mentioned Firmicutes/Bacteroidetes ratio in discussion. However, they did not show their results of F/B ratio. Please show them.
7. The effects of SDexF on mice and children were different. The authors should discuss possible factors of this result.
8. I think that meal or food intake of participants is important to understand the results of this manuscript. If possible, the authors should show the food intake of two groups and discuss the relationship between food intake and the effect of SDexF.
Author Response
Dear Reviewer,
we appreciate the Reviewers' comments. We added our responses below.
This manuscript is reporting the effects of dietary fiber from potato starch on mice and human children. The results were different between mice and children. This manuscript may contribute for the readers to understand the role of prebiotics in obesity. There are some comments.
- The authors showed body weight change during mouse study. I wonder if SDexF supplementation changed the taste of animal foods. The authors should show food intake of eight groups.
We agree that measurements of food intake could answer the above question, but unfortunately these measurements were not conducted systematically throughout the experiment. We have added this information in the revised version of the manuscript.
- The authors showed the result of liver fat with histological assessment. I think that measurement of liver TG content is more objective.
Histopathological examination of the liver is recognized a standard method of measuring the degree of liver steatosis in clinical practice (Kleiner, D.E.; Brunt, E.M.; Van Natta, M.; Behling, C.; Contos, M.J.; Cummings, O.W.; Ferrell, L.D.; Liu, Y.-C.; Torbenson, M.S.; Unalp-Arida, A.; et al. Design and Validation of a Histological Scoring System for Nonalcoholic Fatty Liver Disease. Hepatology 2005, 41, 1313–1321, doi:10.1002/hep.20701) and therefore microscopic examinations, not the measurement of TG content in the liver, were performed in our study.
- In my case, Fig. 2 does not fit on the page.
Corrected.
- Tables shows p-values. However, p-value is not sufficient to understand research results (American Statistical Association). The authors should show their results as raw data, such as values of index and microbial abundance.
We have changed the presentation of the results of metagenomic and metabolomic analyzes in line with the Reviewer's expectations. Because the change in the presentation method significantly increased the volume of data presented, some results previously presented in the form of tables were replaced by presentation in the form of figures.
- Discussion is redundant and vague. Please reconstruct it.
As the Reviewer expected, the discussion of our paper has been significantly changed.
- The authors mentioned Firmicutes/Bacteroidetes ratio in discussion. However, they did not show their results of F/B ratio. Please show them.
Since the results of the F/B ratio did not bring any significant new elements to the work, we removed the mentioned information from the text of the discussion.
- The effects of SDexF on mice and children were different. The authors should discuss possible factors of this result.
We have added the following explanations to the discussion text: “Although it is unclear whether the dose of 10 g of SDexF a day was insufficient to induce weight loss effects in our participants, significant differences in the effects of SDexF observed in mice and children could be due to a higher SDexF content in the di-et of mice than in children. It also cannot be ruled out that SDexF administered to mice in higher concentrations resulted in lower consumption of the diet containing SDexF than that without SDexF. Unfortunately, we had not consistently monitored dietary intake between SDexF-consuming groups and those who did not. Although sporadic evaluations of dietary intake during the 28-week experimental period did not show significant differences”.
- I think that meal or food intake of participants is important to understand the results of this manuscript. If possible, the authors should show the food intake of two groups and discuss the relationship between food intake and the effect of SDexF.
We agree that measuring food intake in groups of children exposed to SDexF and not treated would add additional information to this work. But our research protocol did not include this element, only advice and training on the weight loss procedure itself.

Reviewer 2 Report
Comments and Suggestions for Authors
It is suggested that the paper be modified for certification, and the opinions are as follows.
1. We show that the effects of potato starch soluble dextrin fiber on body weight and related intestinal disorders are evident in mice fed a Western diet, but not in overweight/obese children, and need to explain why; It has no effect on population study. What is the significance of preparing this product in this paper?
2. Please confirm whether the mouse model C57BL6/W is correct. Generally, C57BL/6J and C57BL/6N are used.
3. Why do you use 12-week-old mice, don't you think it's good to use 3-4 week-old mice to correspond to childhood in the population?
4. What is the reason for the choice of dosage for children and mice?
5. Please use a flow chart to show the animal tests more clearly.
6. Table 1 Statistical and significance analysis required?
7. Bacteroidota, Firmicutes, Butyricicoccaceae, Prevotellaceae, Helicobacteraceae, Atopobiaceae, and Sutterellaceae Verrucomicrobiota these phylum and family level bacteria do not need italics.
8. Abundance analysis charts or tables of animal flora should be provided and significance analysis should be performed
9. SCAFs should be changed to fatty acid salts such as butyrate, propionate, andacetate. You can refer to the published paper, e.g. DOI:10.1021/acs.jafc.3c05339
10. The similarities and differences in the regulated flora between animal and human experiments need to be discussed. Are these causes not playing a role in children?
Comments on the Quality of English Language
no
Author Response
Dear Reviewer,
we appreciate the Reviewers' comments. We added our responses below.
It is suggested that the paper be modified for certification, and the opinions are as follows.
- We show that the effects of potato starch soluble dextrin fiber on body weight and related intestinal disorders are evident in mice fed a Western diet, but not in overweight/obese children, and need to explain why; It has no effect on population study. What is the significance of preparing this product in this paper?
As stated in our discussion, “Clinical trials showed that two of the most popular commercially available resistant dextrins, Nutriose, a soluble fiber made from wheat, maize or pea starch, and Fibersol-2, a soluble maltodextrin produced from corn originated starch, have multiple positive effects on various health markers [46]. Both prebiotics have beneficial effects on BMI and weight loss in overweight adults, improve glucose and lipid metabolism, stimulate the production of satiety peptides, improve serum immunologic indicators, and reduce endotoxin and inflammation levels. They also have beneficial effects on the gut microbiota, such as promoting the growth of health-beneficial bacteria and increasing SCFA production [46]”. In addition, “The purpose of our study was to assess whether soluble dextrin fiber (SDexF) produced from potato starch can support the weight loss in children with overweight/obesity who underwent an obesity intervention and/or may reduce the risk of weight gain in mice, and to determine whether these possible effects are accompanied by changes in the composition of the gut microbiome and metabolites”.
- Please confirm whether the mouse model C57BL6/W is correct. Generally, C57BL/6J and C57BL/6N are used.
This designation refers to mice from our own breeding facility.
- Why do you use 12-week-old mice, don't you think it's good to use 3-4 week-old mice to correspond to childhood in the population?
The purpose of our work was not to directly compare the effect of SDexF in animal and human studies, but only “whether soluble dextrin fiber (SDexF) produced from potato starch can support the weight loss in children with overweight/obesity who underwent an obesity intervention and/or may reduce the risk of weight gain in mice”. Furthermore, “the studies were conducted in accordance with the Directives of the European Parliament and of the Council (2010/63/EU) and the Polish regulations on the protection of animals used for scientific and educational purposes (Journal of Laws 2021, items 1331 and 2338). Since 4-6 week old mice are just weaned from their mothers, the use of younger mice in this type of study would not be ethically justified.
- What is the reason for the choice of dosage for children and mice?
In both cases, these were the maximum doses well tolerated by children and mice.
- Please use a flow chart to show the animal tests more clearly.
In fact, it is common for publications to include diagrams outlining the course of an experiment, especially for more intricate experimental procedures. However, in our study, the experimental setup is relatively straightforward. We defer the decision to include such a diagram to the editor, bearing in mind that our manuscript already features ten figures and numerous tables.
- Table 1 Statistical and significance analysis required?
The results of the histological tests were presented in a descriptive manner; statistical analysis of these results was not possible.
- Bacteroidota, Firmicutes, Butyricicoccaceae, Prevotellaceae, Helicobacteraceae, Atopobiaceae, and Sutterellaceae Verrucomicrobiota these phylum and family level bacteria do not need italics.
Most journals require the names of the bacterial taxon to be written in italics. That is why we ask the Editor for the final opinion.
- Abundance analysis charts or tables of animal flora should be provided and significance analysis should be performed
Corrected.
- SCAFs should be changed to fatty acid salts such as butyrate, propionate, andacetate. You can refer to the published paper, e.g. DOI:10.1021/acs.jafc.3c05339
Again, we leave the decision as to the form of presentation of SCFA names to the Editor. In our previously published manuscript (Front. Cell. Infect. Microbiol. 13:1190910. doi: 10.3389/fcimb.2023.1190910) we present SCFAs as acids.
- The similarities and differences in the regulated flora between animal and human experiments need to be discussed. Are these causes not playing a role in children?
The differences between human and mouse microbiomes are evident, yet attributing these variances to the distinct effects of prebiotics in children and mice would be mere conjecture. We aim to refrain from such speculation.

Reviewer 3 Report
Comments and Suggestions for Authors
In the arm human study.....How did authors deal with sarcopenic obesity?
If data are available, they should present them.
Vice versa , authors are requested to stress these three following aspects in the Discussion section.
A recent systematic review of eight studies, found that SO is highly prevalent in children and adolescents and is associated with various adverse health outcomes.The prevalence of SO ranged from 5.66% to 69.7% in girls, with a range between 7.2% and 81.3% in boys....as evident in....Front. Endocrinol., 01 June 2022- Volume 13 - 2022 | https://doi.org/10.3389/fendo.2022.914740.
On the other hand, a limitation in the field of clinical investigation of sarcopenic patients is the lack of a generally accepted definition coupled with the difficulty of adopting common diagnostic criteria. In fact, both do not permit to clarify the exact prevalence rate and consequently limit physicians to establish any kind of therapeutical approach or, when possible, to adopt preventive measures. Unfortunately, there is no standardized cure, apart from doing more physical activity and embracing a balanced diet, but newly discovered substances start being considered...as evident in...Sarcopenia, a condition shared by various diseases: can we alleviate or delay the progression? Intern Emerg Med. 2023 Oct;18(7):1887-1895. doi: 10.1007/s11739-023-03339-z. Epub 2023 Jul 25. PMID: 37490203; PMCID: PMC10543607.
Still, the vital role of the gut microbiome through the gut-muscle axis in sarcopenia is increasingly recognized....as evident in...Metabolites of the gut microbiota may serve as precise diagnostic markers for sarcopenia in the elderly. Front Microbiol. 2023 Dec 21;14:1301805. doi: 10.3389/fmicb.2023.1301805. PMID: 38188577; PMCID: PMC10768011.
Comments on the Quality of English LanguageThe English is acceptable
Author Response
Dear Reviewer,
we appreciate the Reviewers' comments. We added our responses below.
In the arm human study.....How did authors deal with sarcopenic obesity?
If data are available, they should present them.
Vice versa, authors are requested to stress these three following aspects in the Discussion section.
A recent systematic review of eight studies, found that SO is highly prevalent in children and adolescents and is associated with various adverse health outcomes.The prevalence of SO ranged from 5.66% to 69.7% in girls, with a range between 7.2% and 81.3% in boys....as evident in....Front. Endocrinol., 01 June 2022- Volume 13 - 2022 | https://doi.org/10.3389/fendo.2022.914740.
On the other hand, a limitation in the field of clinical investigation of sarcopenic patients is the lack of a generally accepted definition coupled with the difficulty of adopting common diagnostic criteria. In fact, both do not permit to clarify the exact prevalence rate and consequently limit physicians to establish any kind of therapeutical approach or, when possible, to adopt preventive measures. Unfortunately, there is no standardized cure, apart from doing more physical activity and embracing a balanced diet, but newly discovered substances start being considered...as evident in...Sarcopenia, a condition shared by various diseases: can we alleviate or delay the progression? Intern Emerg Med. 2023 Oct;18(7):1887-1895. doi: 10.1007/s11739-023-03339-z. Epub 2023 Jul 25. PMID: 37490203; PMCID: PMC10543607.
Still, the vital role of the gut microbiome through the gut-muscle axis in sarcopenia is increasingly recognized....as evident in...Metabolites of the gut microbiota may serve as precise diagnostic markers for sarcopenia in the elderly. Front Microbiol. 2023 Dec 21;14:1301805. doi: 10.3389/fmicb.2023.1301805. PMID: 38188577; PMCID: PMC10768011.
From the point of view of the pathophysiology of obesity, the analysis of the intestinal microbiome in sarcopenic obesity is indeed an important research issue. But, as the Reviewer himself emphasizes, there is no generally accepted definition of sarcopenic obesity, nor common diagnostic criteria. Additionally, treatment of this type of obesity is based, as in other forms of obesity, on increased physical activity and dietary modifications.
Regarding sarcopenic obesity we compared % of body fat and % of muscle mass between treatment groups, which were similar. At the start of the study, in children randomized to the dextrin and placebo group, the % of body fat was: 26,4% and 26,6%, respectively, whereas % of muscle mass were: 40,6% and 40,5%, respectively. At the end of the treatment, in children treated with dextrin and with placebo, the % of body fat was 26,4% and 26,1%, respectively, whereas % of muscle mass were: 40,6% and 40,8%, respectively. These data were added to the text (lines 392-397). However, we did not perform muscle strength tests, which limits observations on sarcopenic obesity in the study sample.

Round 2
Reviewer 1 Report
Comments and Suggestions for Authors
To the authors
This manuscript has been improved. However, there are points to be considered.
1. The authors showed the result of liver fat with histological assessment. I think that measurement of liver TG content is more objective.
In human studies, researchers can obtain very small specimen from participant’s liver. Therefore, I agree that it is difficult to assess TG content of the liver and histological assessment is chosen to assess steatosis. But in mouse study, I think that researchers can obtain enough amount of liver specimen to measure TG content. So I think that the authors should perform both assays in this study.
2. Th authors replied that “We agree that measuring food intake in groups of children exposed to SDexF and not treated would add additional information to this work. But our research protocol did not include this element, only advice and training on the weight loss procedure itself.” I think that this is a limitation of this study. The authors should describe this in the discussion as a limitation.
3. Discussion has been improved.
4. In discussion, although the authors repeated their results of sex dimorphism, they did not discuss possible reason of this phenomenon. Please provide the discussion.
5. I am interested in the advice provided by the authors. As a result of the advice, did the control group eat more vegetables and fruits than those before the participation of this study? If so, I think that this can affect their study results.
Author Response
Reviewer 1, round 2
Dear Reviewer,
we appreciate the Reviewers' comments. We added our responses below.
This manuscript has been improved. However, there are points to be considered.
- The authors showed the result of liver fat with histological assessment. I think that measurement of liver TG content is more objective.
In human studies, researchers can obtain very small specimen from participant’s liver. Therefore, I agree that it is difficult to assess TG content of the liver and histological assessment is chosen to assess steatosis. But in mouse study, I think that researchers can obtain enough amount of liver specimen to measure TG content. So I think that the authors should perform both assays in this study.
The purpose of our work was not to assess the TG content in the livers of the tested mice, but only to assess whether soluble dextrin fiber (SDexF) produced from potato starch can support weight loss in children with overweight/obesity who underwent an obesity intervention and/or may reduce the risk of weight gain in mice, and to determine whether these possible effects are accompanied by changes in the composition of the gut microbiome and metabolites. Sections taken from mouse livers were fixed in formalin. The fixed tissue would probably be suitable for TG determinations, although we have never conducted such tests in our lab. Since the result of these determinations would not change the final conclusions of our work, we have not decided to develop a method to determine TG in tissue. We hope that this time the Reviewer will accept our point.
- The authors replied that “We agree that measuring food intake in groups of children exposed to SDexF and not treated would add additional information to this work. But our research protocol did not include this element, only advice and training on the weight loss procedure itself.” I think that this is a limitation of this study. The authors should describe this in the discussion as a limitation.
The following information was included in the revised manuscript. “However, our research protocol included qualitative rather than quantitative measurement of food intake by children exposed to SDexF which may be considered a limitation of our work”.
- Discussion has been improved.
- In discussion, although the authors repeated their results of sex dimorphism, they did not discuss possible reason of this phenomenon. Please provide the discussion.
The following paragraph was included in the discussion text:
Gonadal hormones can alter the gut microbiome that has been reported in humans and rodents (as summarized by Nutrients. 2023 May 2;15(9):2175. doi: 10.3390/nu15092175). Significant sex differences in lipid metabolism and microbiota composition at baseline, along with sex-dependent responses to HFD, have recently been reported (Nutrients. 2023 May 2;15(9):2175. doi: 10.3390/nu15092175). Female mice exhibited less body weight gain and body fat composition, accompanied by enriched growth of beneficial microbes (eg Akkermansia) and depleted growth of Adlercreutzia and Enterococcus. As previously reported, sex can have a greater impact on the composition of the gut microbiota composition than HFD and antibiotics (Biol Sex Differ. 2020 Jan 20;11(1):5. doi: 10.1186/s13293-020-0281-3). Furthermore, prebiotics have been shown to alleviate anxiety in female mice by changing the composition of the gut microbiota composition in a sex‐specific manner (CNS Neurosci Ther. 2023 Jun;29 Suppl 1(Suppl 1):115-128. doi: 10.1111/cns.14091), and sexual dimorphism of the axis of "gut microbiota-host metabolism” mainly involves changes in the metabolism of SCFAs and amino acids (Biochim Biophys Acta Mol Basis Dis. 2021 Dec 1;1867(12):166266). However, the final mechanisms by which the gut microbiota can associate with a sexually dimorphic response to an HFD are still unclear.
- I am interested in the advice provided by the authors. As a result of the advice, did the control group eat more vegetables and fruits than those before the participation of this study? If so, I think that this can affect their study results.
The aim of the advice was to change the lifestyle of obese children, including changing the Western-type diet to a more health-promoting diet which resulted in a decrease in body weight in 60% of the participants, but without any further impact of dextrin on the weight loss process. This, of course, was associated with the introduction of larger amounts of vegetables and fruit into the diet,

Reviewer 2 Report
Comments and Suggestions for Authors
The author did not give a reasonable explanation for the problems raised by the reviewers, and the suggested changes were rarely revised. The problems in the article have been given by the reviewer last time. Build the last review. For example, since soluble dextrin fiber from potato starch has no effect on children but only on animals, the research in this paper is of little significance.
Because the mice used are not childhood mice may also be the reason why it does not work. The consumption of 4-6 weeks old mice also conforms to the 3R principle, and the corresponding use significance of children.
Only bacteria at the genus level and below need italics, and the presence of short-chain fatty acid salts in feces is not short-chain fatty acids, and these simple questions should not be revised by the author.
E.g., Table 1 Statistical and significance analysis required?;Please use a flow chart to show the animal tests more clearly;The similarities and differences in the regulated flora between animal and human experiments need to be discussed. Are these causes not playing a role in children? Reasons need to be explained in the manuscript and not replied to the reviewer.
Comments on the Quality of English Languageno
Author Response
Reviewer 2, round 2
Dear Reviewer,
we appreciate the Reviewers' comments. We added our responses below.
- The author did not give a reasonable explanation for the problems raised by the reviewers, and the suggested changes were rarely revised. The problems in the article have been given by the reviewer last time. Build the last review. For example, since soluble dextrin fiber from potato starch has no effect on children but only on animals, the research in this paper is of little significance.
If the Reviewer considers only positive results worthy of interest, consistent with the research hypothesis, it is difficult not to agree that our research is of little importance. However, we believe that negative results that do not confirm previous assumptions should also be published. Even more so when it is not consistent with previously published research, as we already explained in the previous answer, As stated: “Clinical trials showed that two of the most popular commercially available resistant dextrins, Nutriose, a soluble fiber made from wheat, maize or pea starch, and Fibersol-2, a soluble maltodextrin produced from corn originated starch, have multiple positive effects on various health markers [46]. Both prebiotics have beneficial effects on BMI and weight loss in overweight adults, improve glucose and lipid metabolism, stimulate the production of satiety peptides, improve serum immunologic indicators, and reduce endotoxin and inflammation levels. They also have beneficial effects on the gut microbiota, such as promoting the growth of health-beneficial bacteria and increasing SCFA production [46]”.
- Because the mice used are not childhood mice may also be the reason why it does not work. The consumption of 4-6 weeks old mice also conforms to the 3R principle, and the corresponding use significance of children.
As we have already explained, the probable reason for the inconsistency between the results of our studies in children and mice is the too low dose of probiotic administered to the children, and not the age of the mice. However, this was the highest dose of prebiotic well tolerated by children.
We understand that the Reviewer assumes there is no effect or a different effect of the probiotic in younger mice. Unfortunately, it is not possible to repeat this experiment in younger mice due to the lack of funding for repeating the experiment and the lack of consent from our Ethics Committee.
- Only bacteria at the genus level and below need italics, and the presence of short-chain fatty acid salts in feces is not short-chain fatty acids, and these simple questions should not be revised by the author. E.g., Table 1 Statistical and significance analysis required?
We followed the reviewer's recommendations.
- Please use a flow chart to show the animal tests more clearly.
The flow chart was included.
- The similarities and differences in the regulated flora between animal and human experiments need to be discussed. Are these causes not playing a role in children? Reasons need to be explained in the manuscript and not replied to the reviewer.
Despite many years of research on the role of the intestinal microbiome in health and disease, we are unable to determine what human and mouse regulatory mechanisms are intestinal bacteria subject to? In other words, we cannot determine which elements of microbiota changes are a consequence and which are the result of obesity.

Reviewer 3 Report
Comments and Suggestions for Authors
Authors are kindly requested to put in the text the content of the answer regarding sarcopenic obesity with their relevant data.
Author Response
Dear Reviewer,
we appreciate the Reviewers' comments.